# BIGRoC: Boosting Image Generation via a Robust Classifier

**BigGAN output**

**BIGRoC output**

FID: 7.45, IS: 9.38

FID: 6.79, IS: 9.47

Figure 1: **BIGRoC results:** left: images of 3 classes (birds, cars and trucks) generated by BigGAN (Brock et al., 2019), trained on CIFAR-10. Right: the enhanced versions of these images, attained by the proposed BIGRoC algorithm.

## Abstract

The interest of the machine learning community in image synthesis has grown significantly in recent years, with the introduction of a wide range of deep generative models and means for training them. Such machines' ultimate goal is to match the distributions of the given training images and the synthesized ones. In this work, we propose a general model-agnostic technique for improving the image quality and the distribution fidelity of generated images, obtained by any generative model. Our method, termed BIGRoC (boosting image generation via a robust classifier), is based on a post-processing procedure via the guidance of a given robust classifier and without a need for additional training of the generative model. Given a synthesized image, we propose to update it through projected gradient steps over the robust classifier, in an attempt to refine its recognition. We demonstrate this post-processing algorithm on various image synthesis methods and show a significant improvement of the generated images, both quantitatively and qualitatively.

## 1 Introduction

Deep generative models (DGMs) are a class of deep neural networks trained to model complicated high-dimensional data (Bond-Taylor et al., 2021). Such models receive a large number of samples that follow a certain data distribution, $x \sim P_D(x)$, and aim to produce samples from the same statistics. One of the most fascinating generative tasks is image synthesis, which is notoriously hard, due to the complexity of the natural images' manifold. Nevertheless, deep generative models for image synthesis have gained tremendous popularity in recent years, revolutionized the field and became state-of-the-art in various tasks (Brock et al., 2019; Karras et al., 2020b; Zhu et al., 2017). Energy-based models, variational autoencoders, generative adversarial networks (GANs), autoregressive likelihood models, normalization flows, diffusion-based algorithms and more, all aim to synthesize natural-looking images, ranging from relatively simple to extremely complicated generators, often containing millions of parameters (Kingma & Welling, 2014; Goodfellow et al., 2014; Oord et al., 2016; Rezende & Mohamed, 2015; Ho et al., 2020).

When operating on a multiclass labeled dataset, as considered in this paper, image synthesis can be either conditional or unconditional. In the unconditional setup, the generative model aims to

produce samples from the target data distribution without receiving any information regarding the target class of the synthesized images, i.e., sample from $P_D(x)$. In contrast, in the conditional setup, the generator goal is to synthesize images from a designated class, i.e., sample from $P_D(x|y)$ where $y$ is the label. As such, conditional generative models receive additional class-related information.

Most of the work in the deep generative models' field has been focusing on improving the quality and the variety of the images produced by such models, tackled by seeking novel architectures and training procedures. In this work, while still aiming to improve the performance of trained generative models, we place a different emphasis than in most of these studies and propose a method for boosting generative models without any re-training or fine-tuning. More specifically, our method improves the perceptual quality of the images synthesized by any given model via an iterative post-processing procedure driven by a *robust classifier*.

With the introduction of learning-based machines into "real-world" applications, the interest in the robustness of such models has become a central concern. While there are abundant of definitions for robustness, the most common and studied is the adversarial one. This definition upholds if a classifier is robust to a small perturbation of its input, made by an adversary in order to fool it. Previous work (Szegedy et al., 2014) has demonstrated that deep neural networks are not robust at all and can be easily fooled by an adversary. In light of this observation, many robustification methods were proposed, but the most popular among these is adversarial training (Goodfellow et al., 2015). According to this method, in order to train a robust classifier, one should generate adversarial examples and incorporate them into the training process. While examining the properties of such classifiers, researchers have revealed a fascinating phenomenon, called *perceptually aligned gradients* (Tsipras et al., 2019). According to this tendency, a modification of an image that sharpens such a classifier's decision yields visual features that are perceptually aligned with the target class. In other words, when drifting an image content to be better classified, the changes obtained are visually pleasing and faithful to natural image content.

In this work we harness and utilize the above described phenomenon – we propose to iteratively modify the images created by a trained generative model, so as to maximize the conditional probability of a certain target class, approximated by a given robust classifier. This modification can potentially improve the quality of the synthesized images, since it emphasizes visual features that are aligned with images of the target class, thus boosting the generation process both in terms of perceptual quality and distribution faithfulness. We hypothesize that given an image dataset, the supervised training of a robust classifier is much simpler and effective than the unsupervised training of a generative model, thus enabling an indirect yet powerful improvement of generative models. We term this method "BIGRoC" – Boosting Image Generation via a Robust Classifier.

The method presented in this article is general and model-agnostic, and it can be applied to any image generator, both conditional or unconditional. In the unconditional case, since we do not have a target class to guide the boosting process, we propose to estimate it via the trained robust classifier. The marked performance improvement achieved by our proposed method is demonstrated in a series of experiments on a wide range of image generators. We show that this approach enables us to significantly improve the quality of images synthesized by relatively simple models, boosting them to a level of more sophisticated and complex generators. Furthermore, we demonstrate the ability of our method to enhance the performance of higher-quality generative architectures, both qualitatively and quantitatively. In addition to our contribution on enhancing image generation performance, we leverage the same robust classifier and its perceptually aligned gradient property for proposing an adversarial technique for image interpolation.

Our approach is inspired by two main works: Santurkar et al. (2019) and Turner et al. (2019). Santurkar et al. (2019) have shown that a single robust classifier is capable of tackling various computer vision problems, such as generation, inpainting and image-to-image translation, all achieved by utilizing the perceptually aligned gradients phenomenon. Turner et al. (2019) aims at improving the generation quality of GANs, by discarding low quality images, identified by the GAN's discriminator. While our work is related to the above two papers, it differs substantially in the following main characteristics: (i) While our boosting technique relies on the perceptually aligned gradient property as in Santurkar et al. (2019), it builds on the outcome of pre-trained generative models, thus getting

much higher quality images; (ii) Our proposed boosting can operate on any image synthesizer; and (iii) All images produced by the generator are taken into account, none being discarded.

## 2 BACKGROUND

### 2.1 ADVERSARIAL EXAMPLES

Adversarial examples are instances that are intentionally designed by an attacker to cause a false prediction by a machine learning-based classifier (Szegedy et al., 2014). The generation procedure of such examples is based on applying modifications to given training examples, while restricting the allowed perturbations $\Delta$. Ideally, the "threat model" $\Delta$ should include all the possible perturbations that are unnoticeable to a human observer. As it is impossible to rigorously define such a set, in practice a simple subset of the ideal threat model is used, where the most common choices are the $\ell_2$ and the $\ell_\infty$ balls: $\Delta = \{\delta \ : \ \|\delta\|_{2/\infty} \leq \epsilon\}$. Given $\Delta$, the attacker receives an instance $x$ and generates $\hat{x} = x + \delta \ s.t. \ \delta \in \Delta$, while aiming to fool the classifier. Adversarial attacks can be both untargeted or targeted: An untargeted attack perturbs the input in a way that minimizes $p(y|\hat{x})$ with respect to $\delta$. In contrast, a targeted attack receives in addition the target class $\hat{y}$, and perturbs $x$ to maximize $p_(\hat{y}|\hat{x})$. There are diverse techniques for generating adversarial examples, yet, in this work, we focus on targeted attacks using the Projected Gradient Descent (PGD) method (Madry et al., 2018)– an iterative method for creating adversarial examples that operates as shown in Algorithm 1.

---

**Algorithm 1:** Targeted Projected Gradient Descent (PGD) for adversarial targeted attacks

---
**Input**: classifier $f_\theta$, input $x$, target class $\hat{y}$, $\epsilon$, step size $\alpha$, number of iterations $T$
$\delta_0 \leftarrow 0$
**for** *t from 0 to T* **do**
$\quad | \quad \delta_{t+1} = \Pi_\epsilon(\delta_t - \alpha\nabla_\delta\ell(f_\theta(x + \delta_t), \hat{y}));$
**end**
$x_{adv} = x + \delta_T$
**Output**: $x_{adv}$

---

The operation $\Pi_\epsilon$ stands for a projection operator onto $\Delta$, and $\ell(\cdot)$ is the classification loss.

### 2.2 ADVERSARIAL ROBUSTNESS

Adversarial robustness is a property of classifiers, according to which, applying small perturbation on a classifier's input in order to fool it does not affect its prediction (Goodfellow et al., 2015). To attain such classifiers, one should solve the following optimization problem:

$$\min_\theta \sum_{x,y \in D} \max_{\delta \in \Delta} \ell(f_\theta(x + \delta), y) \tag{1}$$

Namely, train the classifier to accurately predict the class labels of the "toughest" perturbed images, allowed by the threat model $\Delta$. In practice, solving this optimization problem is challenging, and there are several ways to attain an approximated solution. The most simple yet effective method is based on approximating the solution of the inner-maximization via adversarial attacks, such as PGD. According to this strategy, the above optimization is performed iteratively, fixing the classifier's parameters $\theta$ and optimizing the attacks $\delta$ for each example via PGD, and then fixing these and updating $\theta$. Repeating these steps results in a robust classifier, as we use in this work.

### 2.3 PERCEPTUALLY ALIGNED GRADIENTS

Perceptually aligned gradients is a phenomenon that occurs in adversarially trained models when modifying an image to maximize the probability assigned to a target class. Tsipras et al. (2019) show that performing the above PGD process on such models yields meaningful visual features that are

perceptually aligned to the target class. It is important to note that this phenomenon does not occur in non-robust models. The perceptually aligned gradients property indicates that the features learned by robust models are more aligned with human perception. Figure 2 presents a visual demonstration of this fascinating phenomenon.

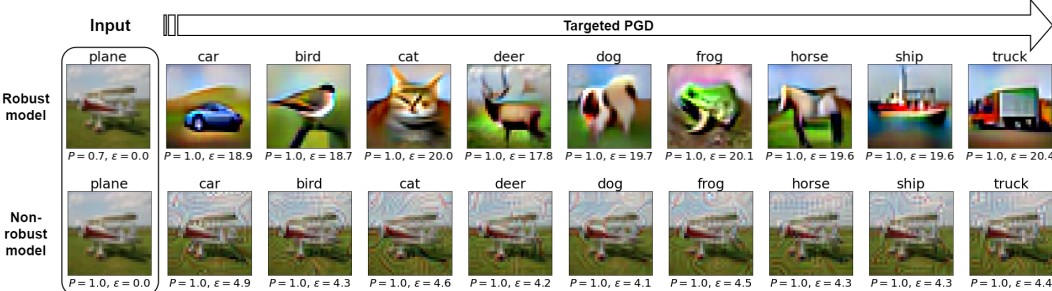

Figure 2: Demonstration of the perceptually aligned gradients phenomenon on CIFAR-10: the original image (left) is perturbed via a targeted PGD attack to maximize the probability of different target classes. This experiment is done with a robust and a non-robust classifiers of the same architecture - ResNet50 (H. et al., 2016). The top row shows the results for the robust classifier and the bottom for the non-robust one. We specify below each image the relevant classifier's certainty (denoted as P), and the effective $\ell_2$ norm of the perturbation $\delta$ (denoted as $\epsilon$). For each of the generated images, the attacked classifier reaches absolute certainty ($P = 1$). As can be seen, an adversarial attack (PGD) on a robust classifier leads to a significant addition of perceptual features aligned with the target class. This phenomenon does not occur at all on the non-robust classifier. Although both networks use the same threat model $\Delta$, we see that the attack on the non-robust classifier does not utilize the entire threat model and perturbs the image in a seemingly insignificant way, but still able to completely fool it. Experimental settings: $\Delta = \{\delta \; : \; \|\delta\|_2 \leq \epsilon\}$, $\epsilon = 30$, $T = 60$, $\alpha = \frac{\epsilon}{T} = 0.5$.

## 3 BOOSTING IMAGE GENERATION VIA A ROBUST CLASSIFIER

We propose a method for improving the quality of images synthesized by trained generative models, named BIGRoC: Boosting Image Generation via a Robust Classifier. Our method is model agnostic and does not require additional training or fine-tuning of the generative model, and can be viewed as a post processing step. Thus, BIGRoC can be easily applied to any generative model, both conditional or unconditional. This mechanism harnesses the perceptually aligned gradients phenomenon to further tweak the generated images to improve their visual quality. To do so, we perform an iterative process of modifying the generated image $x$ to maximize the posterior probability of a given target class $\hat{y}$, $p_\theta(\hat{y}|x)$, where $p_\theta$ is modeled by an adversarially trained classifier. This can be achieved by performing a PGD-like process, but instead of adversarially changing an image $x$ of class $y$ to a different class $\hat{y} \neq y$, we propose to modify it in a way that maximizes the probability that $x$ belongs to $y$. Therefore, our method requires a trained robust classifier that operates on the same data source as the generator we aim to improve.

In the conditional generation process, the generator $G$ receives the class label $y$, from which it suppose to draw samples. Hence, in this setup, we have information regarding the class affiliation of the image and we can maximize the corresponding conditional probability. In the unconditional generation process, the generator does not receive class labels at all and its goal is to draw samples from $p(x)$. Thus, in this case, we cannot directly maximize the desired posterior probability, as our method suggests. To bridge this gap, we propose to estimate the most likely class via our robust classifier $f_\theta$, and afterward modify the image via the suggested method to maximize its probability. The proposed image generation boosting is described in Algorithm 2, for both the conditional and the unconditional schemes.

While the above-described approach for unconditional sampling works well, it could be further improved. We have noticed that in this case, estimating the target classes $Y$ of $X_{gen}$ via $f_\theta$ leads

to unbalanced labels. For example, in the CIFAR-10, when generating 50,000 samples, we expect approximately 5,000 images per each of the 10 classes, and yet the labels' estimation do not distribute uniformly at all. This imbalance causes a bias in the target labels estimation of the boosting algorithm, affects the visual content of $X_{boost}$ and limits the quantitative improvement attained by BIGRoC. We emphasize that this issue is manifested only in the quantitative metrics, and when qualitatively evaluating the boosted images, the improvement is significant, as can be seen in Figure 3.

To further enhance the quantitative results of our algorithm in the unconditional case, we propose to de-bias the target class estimation of $X_{gen}$, and attain close to uniform class estimations. A naive solution to this can be achieved by generating more samples and extracting a subset of these images with a labels-balance. This approach is computationally heavy and does not use the generated images as-is, which raises questions regarding the fairness of the quantitative comparison. Thus, we propose a different debiasing technique – we modify the classifier's class estimation to become more balanced by calibrating its logits. More specifically, we shift the classifier's logits by adding a per-class pre-calculated value, $d_{c_i}$, that induces equality of the mean logits value across all classes. For simplicity, we denote $logit_{c_i}$ as the logit of class $c_i$ corresponding to a generated sample $x_{gen}$. We approximate $\mathbb{E}_{x_{gen}}[logit_{c_i}]$ for each class $c_i$, using a validation set of generated images, and calculate a per-class debiasing factor: $d_{c_i} = a - \hat{\mathbb{E}}_{x_{gen}}[logit_{c_i}]$ (WLOG, $a = 1$), where $\hat{\mathbb{E}}_{x_{gen}}[logit_{c_i}]$ is a mean estimator. After calculating $d_{c_i}$, given a generated image $x_{gen}$, we calculate its logits and add $d_{c_i}$ to it to obtain debiased logits ($\hat{logit}_{c_i}$), from which we derive the unbiased class estimation via softmax. The following equation shows that, given a correct estimation of the per-class logits' mean, the per-class means of the debiased logits are equal:

$$\mathbb{E}_{x_{gen}}[\hat{logit}_{c_i}] = \mathbb{E}_{x_{gen}}[d_{c_i} + logit_{c_i}] = \mathbb{E}_{x_{gen}}[a - \hat{\mathbb{E}}_{x_{gen}}[logit_{c_i}] + logit_{c_i}] =$$
$$= a - \hat{\mathbb{E}}_{x_{gen}}[logit_{c_i}] + \mathbb{E}_{x_{gen}}[logit_{c_i}] \approx a \qquad (2)$$

Figure 3 presents a demonstration that verifies the validity of this method and shows its qualitative effects on the unconditional generation boosting.

---

**Algorithm 2:** BIGRoC: Boosting Image Generation via a Robust Classifier

---

**Input**: Robust classifier $f_\theta$, $x_{gen}$, $y_{gen}$, $\epsilon$, step size $\alpha$, number of iterations $T$, $d_c$
**if** $y_{gen}$ *is None* **then**
    | $y_{gen} = argmax(f_\theta(x_{gen}) + d_c)$
**end**
$x_{boost} = Targeted\ PGD(f_\theta, x_{gen}, y_{gen}, \epsilon, \alpha, T)$
**Output**: $x_{boost}$

---

As can be seen in Algorithm 2, it receives as input the generated images and their designated labels (if exist) and returns an improved version of them. As such, this method can be applied at the inference phase of generative models to enhance their performance, in a manner totally separated from their training. As can be seen from the algorithm's description, it has several hyperparameters that determine the modification process of the image: $\epsilon$ sets the maximal size of the perturbation allowed by the threat model $\Delta$, $\alpha$ controls the step size at each update step and $T$ is the number of updates. Another choice is the norm used to define the threat model $\Delta$.

The hyperparameter $\epsilon$ is central in our scheme - when $\epsilon$ is too large, the method overrides the input and modifies the original content in an unrecognizable way, as can be seen in Figure 2. On the other hand, when $\epsilon$ is too small, the boosted images remain very similar to the input ones, leading to a minor enhancement. As our goal is to obtain a significant enhancement to the synthesized images, a careful choice of $\epsilon$ should be practiced, which restricts the allowed perturbations in the threat model.

Another important choice is the threat model $\Delta$ itself. Two of the most common choices of $\Delta$ for adversarial attacks are the $\ell_\infty$ and the $\ell_2$ balls. Due to the desired behavior of our method, using the $\ell_\infty$ ball is less preferable: it allows a change of $\pm\epsilon$ to every pixel and as such it will not focus on meaningful specific locations and might not preserve the existing structure of the synthesized input image. Thus, we choose the $\ell_2$ ball as our threat model, with relatively small $\epsilon$. Such a choice restricts the allowed perturbations and leads to changes that may concentrate in specific locations,

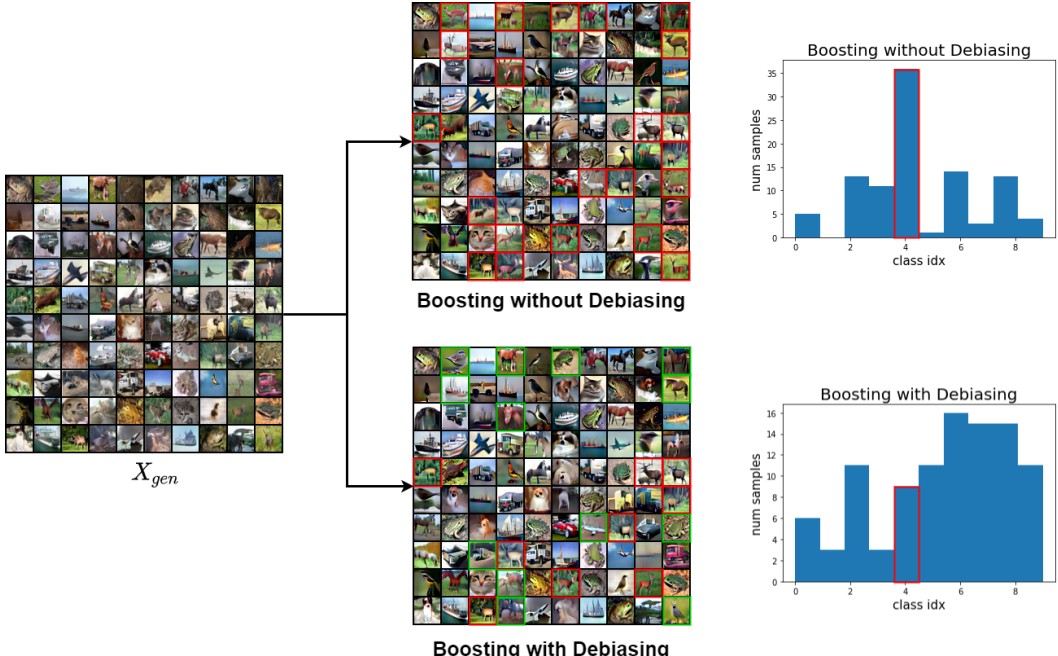

Figure 3: Demonstration of the debiasing technique: We show 100 generated images by an unconditional SNGAN and the results of the BIGRoC algorithm, with and without the proposed debiasing. As can be seen, the outputs of the boosting algorithms are perceptually superior, while the histograms expose the fact that the suggested debiasing algorithm induces a more uniform labels' distribution. In the "Boosting without Debiasing" experiment, 36 out of 100 images are classified as deers, and only 3 are horses. The most prominent deer images are marked in red. However, when applying the debiased boosting, the number of deers is reduced to 9, and the number of horses is increased to 15. We mark the boosted images that remain deer in red, and images that are modified to other minority classes in green. As can be seen, many of the deers were changed to be horses, a perceptually similar class.

while preserving most of the existing content in the generated images. A visual demonstration of these considerations is given in Figure 4.

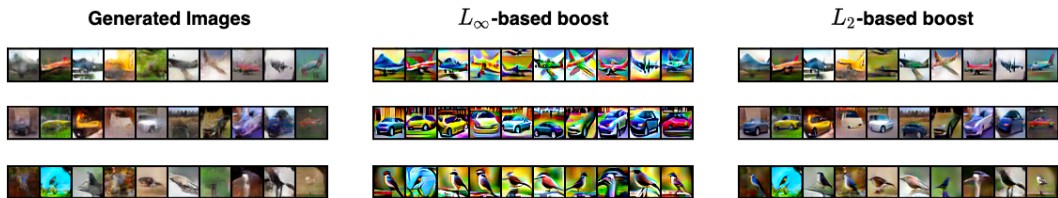

Figure 4: The effect of the threat model's choice: generated images via a conditional GAN (left) are perturbed via a targeted PGD attack to maximize the probability of their target classes (planes, cars and birds) using either $\ell_\infty$ or $\ell_2$ threat models. The boosted outputs attained by the $\ell_\infty$ entirely change the structure of the images and lead to unnatural results. Using the $\ell_2$ threat model leads to pleasing and better results, as claimed. `Experimental settings:` In the $\ell_2$ case, we use $\epsilon = 10$, $T = 30$, $\alpha = \frac{\epsilon}{T} = \frac{1}{3}$, which leads to good visual results. In order to make a sensible comparison with the $\ell_\infty$, we measure the perturbation of each pixel made by the $\ell_2$-based boosting algorithm and calculate the $70\%$ percentile of the perturbations, denoted as $q_{0.7}$. Namely, $q_{0.7}$ is larger than 70% of the perturbations done by our method using the $\ell_2$ threat model. For the $\ell_\infty$ case, we set $\epsilon$ to be equal to the $q_{0.7}$, while leaving the other hyperparameters unchanged.

## 4 ROBUST CLASSIFIER BASED IMAGE INTERPOLATION

As a byproduct of the boosting algorithm for improving the quality of synthesized images, we develop a method for image interpolation, which stems from a similar rationale. In image synthesis, image interpolation is an operation that aims to generate intermediate pictures between two given images, while remaining meaningful and of gradual change. We propose to use features extracted by our robust classifier at different scales in order to guide a PGD-based interpolation process. More specifically, given a source image $x_s$ and a target image $x_t$, we extract their corresponding features $\{F_{x_s}^1, \ldots, F_{x_s}^k\}$ and $\{F_{x_t}^1, \ldots, F_{x_t}^k\}$ from our trained robust classifier $f_\theta$. To perform interpolation, we modify $x_s$ such that its features will be more similar to those of $x_t$ at the different scales, using PGD. Mathematically, we aim to solve Equation 3:

$$arg\,min_{\delta \in \Delta} \sum_{i=1}^{k} w_i \cdot \|F_{x_s+\delta}^i - F_{x_t}^i\|_2 \qquad (3)$$

where $\{w_i\}_{i=1}^k$ is an importance vector to the different features' scales. Namely, this optimization aims to find a perturbation $\delta \in \Delta$ that minimizes the $\ell_2$-distance between the images' representations. In the above formulation, the choice of the $\epsilon$ determines the level of change allowed in the process. Thus, solving the above optimization with different $\epsilon$ values leads to different intermediary images – as we increase $\epsilon$, the interpolation resembles the target image more.

## 5 EXPERIMENTS

**Image Generation Enhancement:** We conduct extensive experiments to demonstrate the performance improvement achieved by the proposed method. In this work, we focus on CIFAR-10 (Krizhevsky, 2012), the most common benchmark for image synthesis. Since BIGRoC is model-agnostic, it can be easily applied to any generative model, given a robust classifier, trained on the same data source. Although our proposed method is a general technique, we experiment mainly with Generative Adversarial Networks (GANs), due to their popularity and generation quality. In all of our experiments, we use a single adversarially trained ResNet-50 (H. et al., 2016) as the robust classifier. We further elaborate regarding the experimental setups in Appendix .2. We utilize the model-agnostic property to examine the effects of applying the proposed boosting over a wide variety of image generators of different qualities: from relatively simple ones to sophisticated and complex ones. Moreover, we test our approach on both conditional and unconditional generative models to show that the proposed scheme can enhance different synthesis procedures.

Given a generative model, we use it to synthesize a set of images $X_{gen}$ and apply our method to generate $X_{boost}$, according to Algorithm 2. We compare between $X_g$ and $X_{boost}$ quantitatively using two image synthesis evaluation methods: Fréchet Inception Distance (FID) (Heusel et al., 2017) and Inception Score (IS) (Salimans et al., 2016). Our quantitative results (Tables 1, 2) indicate that our method achieves a substantial improvement across a wide range of tested generator architectures, both conditional and unconditional, demonstrating the method's versatility and validity. In addition, the fact the same robust classifier is capable of enhancing the performance of such a wide range of generative models, demonstrates the power of BIGRoC. Furthermore, we show in Figure 1, 5 and in Appendix .1 the qualitative results that indicate that the "boosted" results are more pleasing and clear to human observers. All the conducted experiments verify the significant enhancement attained by our method, both quantitatively and qualitatively.

**Image Interpolation:** We conduct image interpolation experiments according to the technique proposed in Section 4. We use the ImageNet dataset (Deng et al., 2009) with resolution of $224 \times 224$ and a ResNet-50 classifier. We compare the visual results attained by our interpolation method using both robust and non-robust classifiers of the same architecture. As can be seen in Figure 6, the interpolation results using a robust classifier are much more convincing and the intermediary images are perceptually plausible. In contrast, the interpolation using a non-robust model leads to much inferior results, and the final interpolation does not look like the target image at all.

Table 1: Conditional BIGRoC quantitative results on CIFAR-10, using FID (lower is better) and IS (higher is better).

| Architecture | FID 10K | FID 50K | IS 50K |
|---|---|---|---|
| cGAN (Mirza & Osindero, 2014) | $33.28 \pm 0.31$ | $29.26 \pm 0.10$ | $6.95 \pm 0.03$ |
| Boosted cGAN | $23.62 \pm 0.28$ | $19.54 \pm 0.02$ | $8.18 \pm 0.04$ |
| Improvement | ↓ 29.03% | ↓ 33.22% | ↑ 17.70% |
| cGAN-PD (Miyato & Koyama, 2018) | $15.01 \pm 0.14$ | $11.03 \pm 0.08$ | $8.36 \pm 0.05$ |
| Boosted cGAN-PD | $12.99 \pm 0.11$ | $8.89 \pm 0.05$ | $8.57 \pm 0.05$ |
| Improvement | ↓ 13.46% | ↓ 19.40% | ↑ 2.51% |
| BigGAN (Brock et al., 2019) | $11.57 \pm 0.09$ | $7.45 \pm 0.08$ | $9.38 \pm 0.05$ |
| Boosted BigGAN | $10.80 \pm 0.05$ | $6.79 \pm 0.02$ | $9.47 \pm 0.02$ |
| Improvement | ↓ 6.66% | ↓ 8.86% | ↑ 0.96% |
| Diff BigGAN (Zhao et al., 2020) | $8.53 \pm 0.094$ | $4.37 \pm 0.03$ | $9.48 \pm 0.03$ |
| Boosted Diff BigGAN | $8.03 \pm 0.043$ | $3.95 \pm 0.02$ | $9.61 \pm 0.03$ |
| Improvement | ↓ 12.45% | ↓ 9.61% | ↑ 1.37% |
| StyleGAN2 ADA (Karras et al., 2020a) | $6.60 \pm 0.04$ | $2.44 \pm 0.01$ | $10.02 \pm 0.04$ |
| Boosted StyleGAN2 ADA | $6.56 \pm 0.04$ | $2.41 \pm 0.01$ | $10.07 \pm 0.04$ |
| Improvement | ↓ 0.61% | ↓ 1.23% | ↑ 0.50% |

Table 2: Unconditional BIGRoC quantitative results on CIFAR-10, using FID (lower is better) and IS (higher is better).

| Architecture | FID 10K | FID 50K | IS 50K |
|---|---|---|---|
| VAE (Kingma & Welling, 2014) | $155.05 \pm 0.55$ | $152.04 \pm 0.19$ | $3.05 \pm 0.01$ |
| Boosted VAE | $90.93 \pm 0.60$ | $88.68 \pm 0.37$ | $6.27 \pm 0.04$ |
| Improvement | ↓ 41.36% | ↓ 42.99% | ↑ 51.16% |
| DCGAN (Radford et al., 2015) | $42.44 \pm 0.06$ | $38.34 \pm 0.11$ | $6.10 \pm 0.01$ |
| Boosted DCGAN | $33.66 \pm 0.26$ | $29.93 \pm 0.05$ | $7.28 \pm 0.04$ |
| Improvement | ↓ 29.48% | ↓ 21.94% | ↑ 19.34% |
| WGAN-GP (Salimans et al., 2016) | $26.59 \pm 0.09$ | $22.62 \pm 0.09$ | $7.49 \pm 0.03$ |
| Boosted WGAN-GP | $20.16 \pm 0.22$ | $16.28 \pm 0.08$ | $8.15 \pm 0.03$ |
| Improvement | ↓ 24.18% | ↓ 28.03% | ↑ 8.81% |
| SNGAN (Miyato et al., 2018) | $21.33 \pm 0.38$ | $17.19 \pm 0.07$ | $8.04 \pm 0.02$ |
| Boosted SNGAN | $17.09 \pm 0.20$ | $13.25 \pm 0.10$ | $8.61 \pm 0.04$ |
| Improvement | ↓ 19.88% | ↓ 22.92% | ↑ 7.09% |
| InfoMaxGAN (Lee et al., 2021) | $19.65 \pm 0.20$ | $15.41 \pm 0.12$ | $8.09 \pm 0.05$ |
| Boosted SSGAN | $15.49 \pm 0.23$ | $11.27 \pm 0.11$ | $8.48 \pm 0.03$ |
| Improvement | ↓ 21.15% | ↓ 26.82% | ↑ 4.85% |
| SSGAN (Chen et al., 2019) | $19.24 \pm 0.17$ | $15.05 \pm 0.06$ | $8.20 \pm 0.02$ |
| Boosted SSGAN | $14.85 \pm 0.16$ | $10.77 \pm 0.06$ | $8.61 \pm 0.03$ |
| Improvement | ↓ 22.82% | ↓ 28.44% | ↑ 5.00% |

## 6 DISCUSSION AND CONCLUSIONS

In this work, we propose a novel method that leverages the perceptually aligned gradients phenomenon for enhancing the visual quality of synthesized images. Our approach does not require additional training of the generative model and it is completely model agnostic. Thus, it can be applied during the inference phase of any model. Clearly, training a robust classifier is a simpler task than training a generative model. Due to the core ability of such a robust classifier to better capture significant visual features, it is capable of effectively and efficiently improving the output of generative models. In a line of experiments, we confirm this rationale and show that this method is highly effective and capable of substantially enhancing both the qualitative and quantitative results of a wide range of generative models. In addition, we develop an image interpolation method that stems from a similar origin – adversarial attacks on a robust classifier.

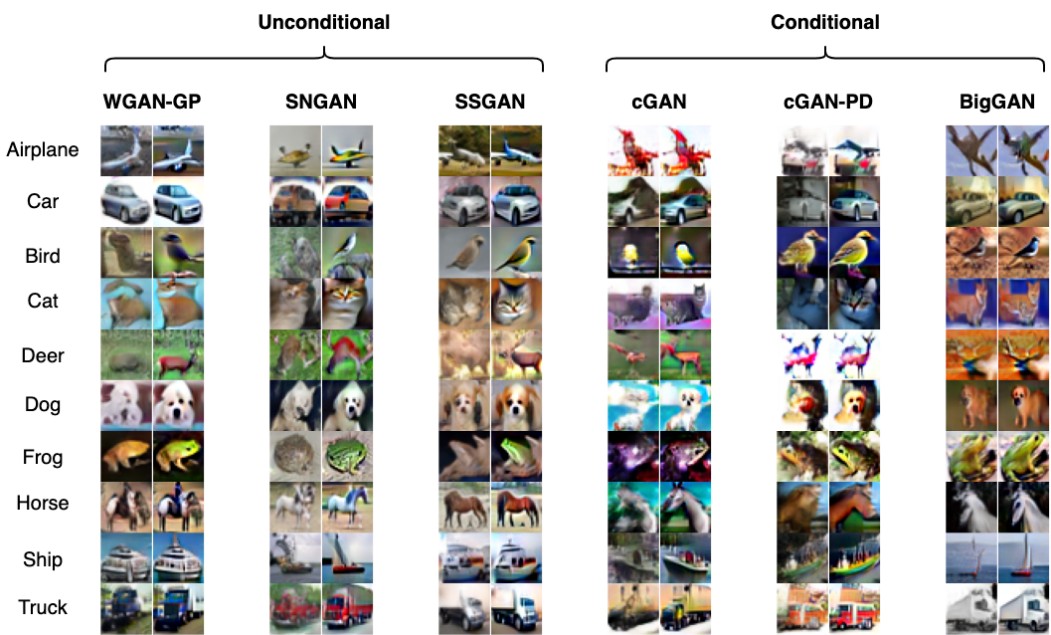

Figure 5: The qualitative enhancement obtained by BIGRoC: We show 6 pairs of sets of images, each containing 10 images, one per CIFAR-10 class. Each set consists of images generated by a certain GAN (left) and their BIGRoC boosted version (right). This demonstrates the significant visual enhancement achieved, both in conditional and unconditional image generation schemes.

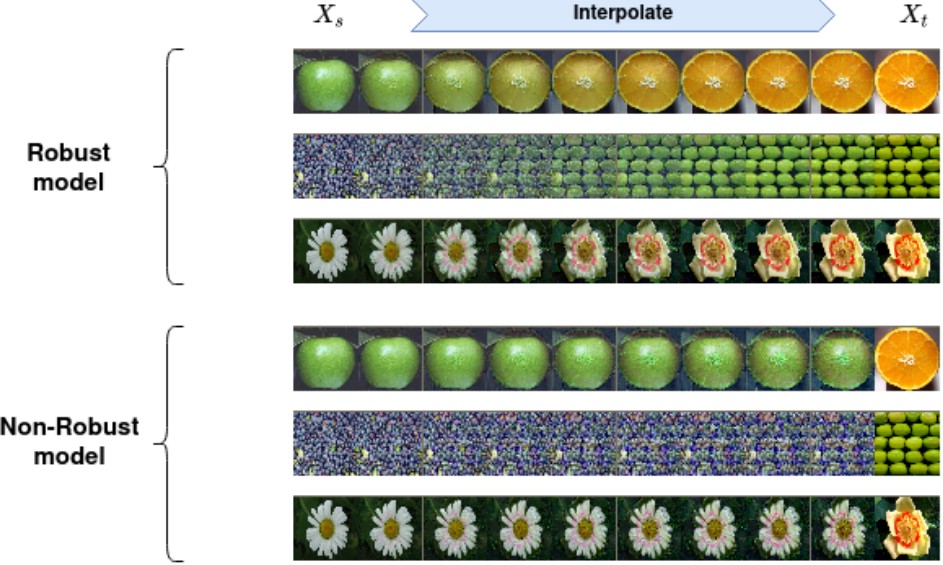

Figure 6: Image interpolation demonstration on ImageNet images. The interpolation process is achieved by an alternation of $\epsilon$ according to $\epsilon = c \cdot \|x_t - x_s\|_2, c \in (0, 1]$.

## 7    REPRODUCIBILITY STATEMENT

In this work we have proposed a post-processing algorithm for enhancing the visual quality of generated images synthesized by generative models. We implemented our approach using the official software or using implementations from Lee & Town (2020) in each of the papers we have referred to, and using the same hyperparameters as stated in their experiments. In addition, in all of our tests,

we evaluated the performance multiple times using different seeds, so as to factor out randomness in the attained improvement. A code for running the conducted experiments for the unconditional generation case is provided in the supplementary materials. A complete code package will be published upon acceptance of the paper.

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

APPENDIX

### .1 ADVERSARIAL IMAGE GENERATION BOOSTING QUALITATIVE RESULTS

In this section, we show additional visual results that further demonstrate the qualitative enhancement attained by our proposed method. We use image generators of different qualities, both conditional and unconditional, and show the generated images and the boosted ones in Figures 7, 8, 9 and 10. All the images shown below are simply the 100 first synthesized images from each class.

### .2 ADVERSARIAL IMAGE GENERATION BOOSTING EXPERIMENTAL SETUP

We use an adversarially trained (Madry et al., 2018) ResNet-50 architecture with an $\ell_2$-based threat model for both image boosting and image interpolation, as in Santurkar et al. (2019). For image boosting, we use the same robust classifier to enhance all tested generative models, and we do not perform any hyperparameter tuning to it. To best harness the perceptually aligned gradients, we use an $\ell_2$ based threat model for the BIGRoC. Since different generative models require different magnitudes of enhancement, we tune the hyperparameter $\epsilon$ in Algorithm 2 to enable the proper enhancement to each of them. The used values of $\epsilon$ across different models are summarized in Table 3. Beyond the value of $\epsilon$, we use the same number of steps (T=30) and set the step size $\alpha$ to be $\frac{\epsilon}{T}$.

Table 3: The hyperparameter $\epsilon$ for BIGRoC for every tested generative model.

| Architecture | $\epsilon$ |
|---|---|
| cGAN | 5 |
| cGAN-PD | 2 |
| BigGAN | 1 |
| Diff BigGAN | 1 |
| StyleGAN2 ADA | 0.28 |
| VAE | 25 |
| DCGAN | 5 |
| WGAN-GP | 5 |
| SNGAN | 5 |
| SSGAN | 3 |

### .3 IMAGE INTERPOLATION ADDITIONAL RESULTS

We show in Figure 11 additional image interpolation results, for both robust and non-robust image classifiers.

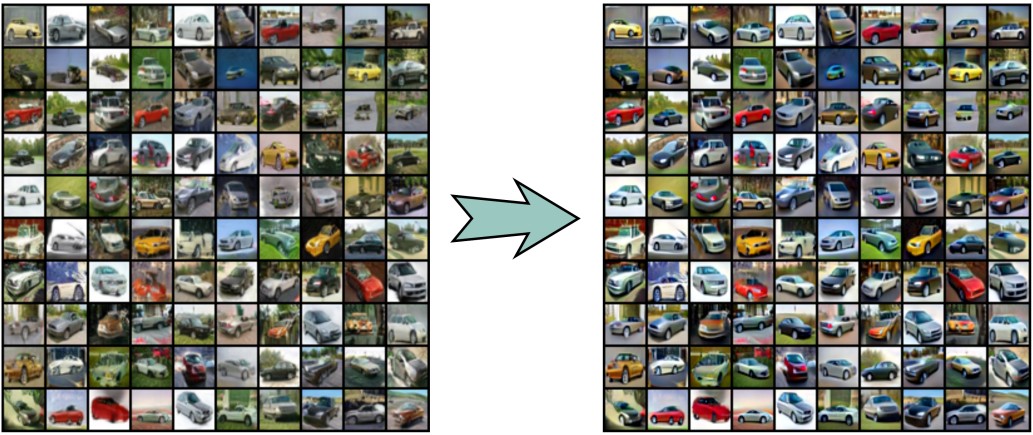

(a) A comparison between BigGAN generated images of class automobile and the corresponding boosted ones.

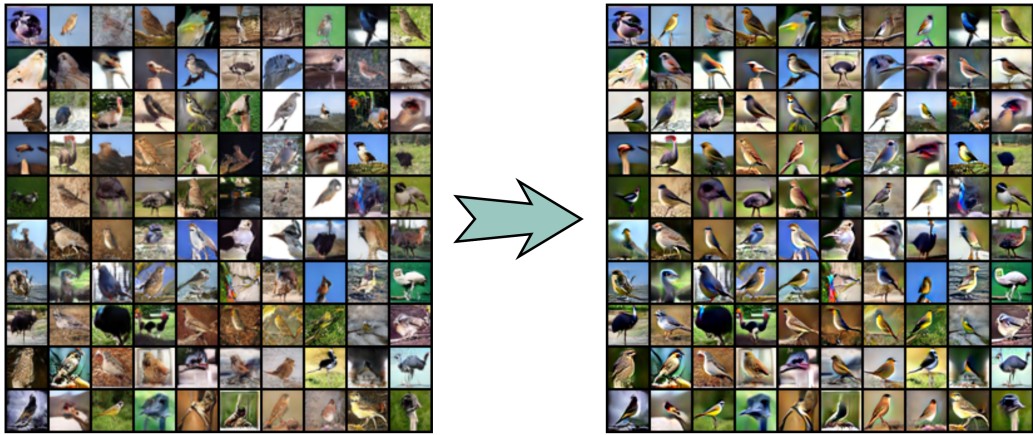

(b) A comparison between BigGAN generated images of class bird and the corresponding boosted ones.

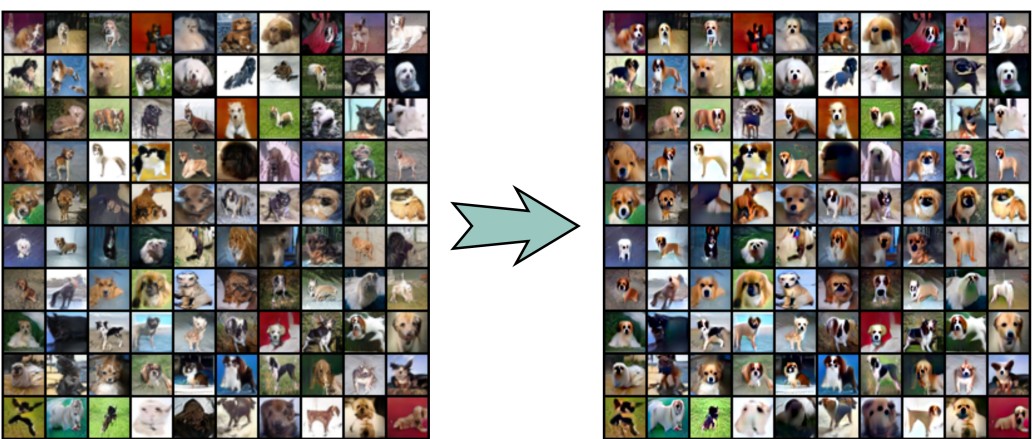

(c) A comparison between BigGAN generated images of class dog and the corresponding boosted ones.

Figure 7: A qualitative comparison between BigGAN generated images of CIFAR-10 samples and the proposed BIGRoC algorithm.

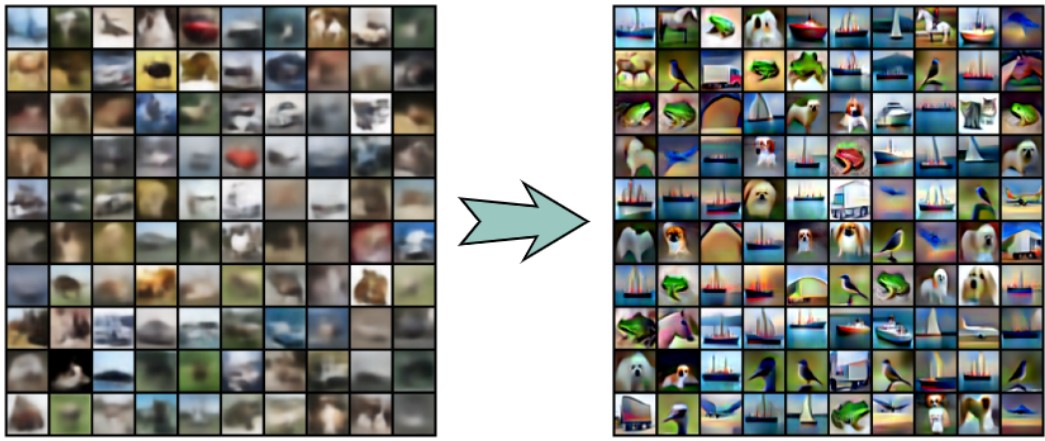

Figure 8: A qualitative comparison between an unconditional VAE generated images of CIFAR-10 samples and the proposed BIGRoC algorithm.

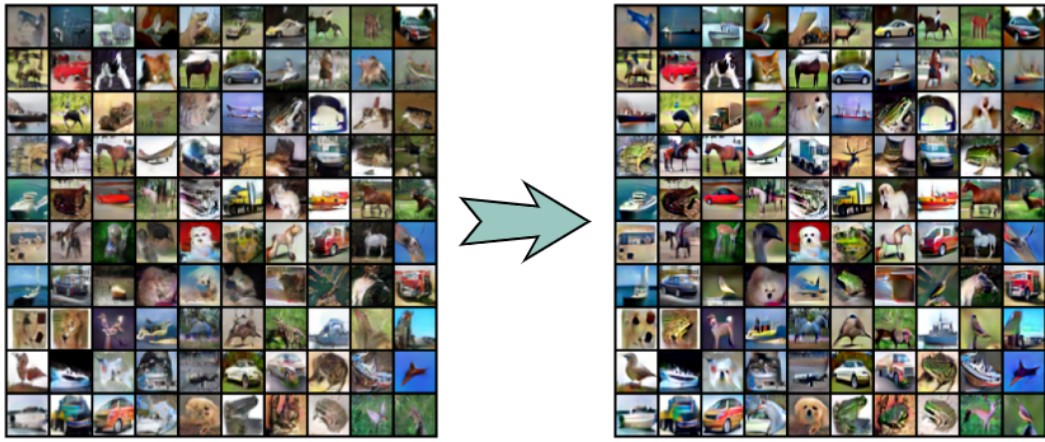

Figure 9: A qualitative comparison between an unconditional WGAN-GP generated images of CIFAR-10 samples and the proposed BIGRoC algorithm.

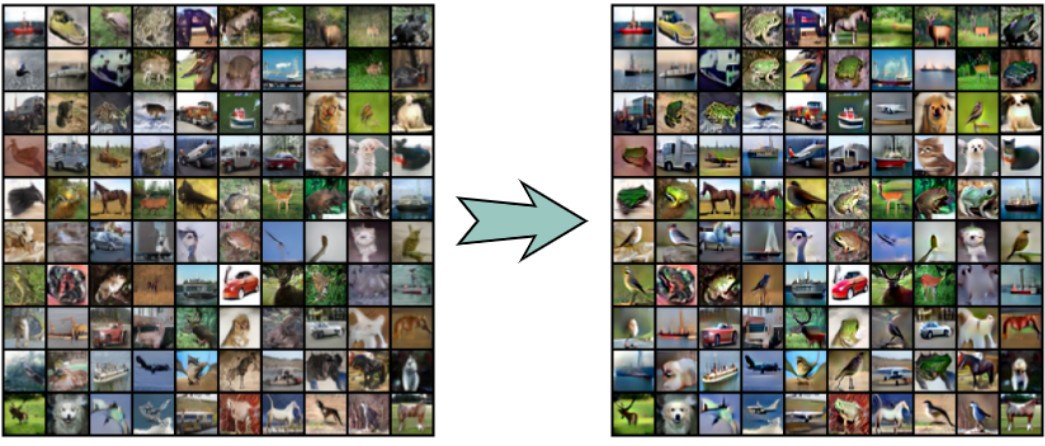

Figure 10: A qualitative comparison between an unconditional SSGAN generated images of CIFAR-10 samples and the proposed BIGRoC algorithm.

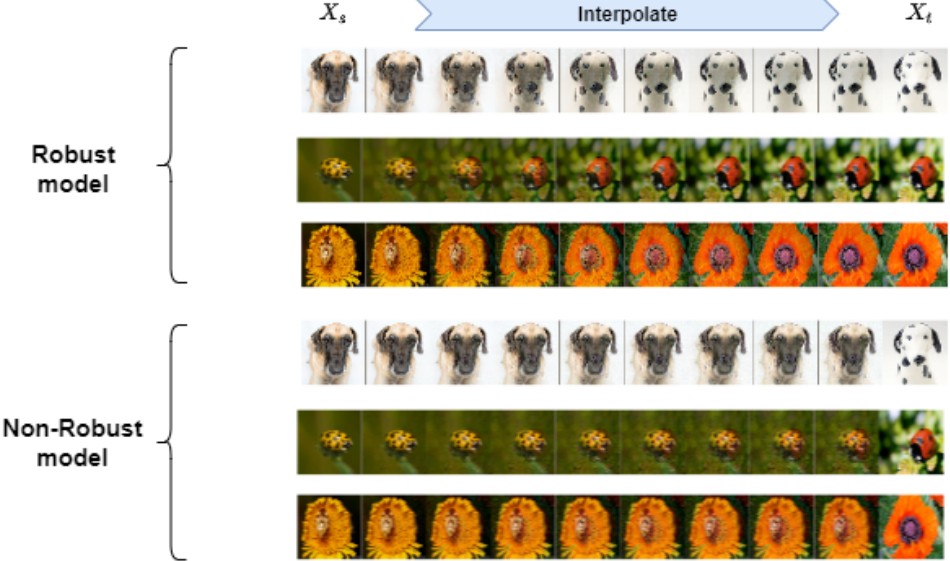

Figure 11: Additional image interpolation results using robust and non-robust classifiers according to our proposed method.

