# OpenReview forum: "BIGRoC: Boosting Image Generation via a Robust Classifier"
_ICLR.cc/2022/Conference — ICLR 2022 Submitted_

### Official Review · Reviewer_Ezr2 · 2021-11-01

**Correctness:** 3
**Technical Novelty And Significance:** 2
**Empirical Novelty And Significance:** 2
**Recommendation:** 3
**Confidence:** 5

**Main Review:**

[Writing]

The manuscript is well-written and easy to understand.

[Novelty, Significance, and Prior Work]

The authors have explored an interesting idea of using robust classifiers for sample refinement. However, the key technical contribution is minor compared to Santurkar et al. [1] — it only involves using a robust classifier on pre-trained generative models. One might argue for the novelty of application, however, the authors have ignored several works that address the problem of sample refinement, e.g., using GAN discriminators [2, 3, 4]. In contrast to these works, BIGRoC also has the additional overhead of training a robust classifier. Moreover, these prior works bring in new technical perspectives to the problem of sample refinement (e.g., from optimal transport [2], energy-based modeling [3], gradient flows [4]) unlike the proposed method which hardly brings in any new technical insights.

[Empirical Evaluation]

It is appreciable that the experiments have been conducted on several base models. However, only the CIFAR-10 dataset has been studied. It is unclear:

- How the method will perform on other datasets?
- How sensitive is the method to the training of the robust classifier?
- What is the additional time overhead of training a robust classifier?

Furthermore, the method has not been evaluated against any baseline methods (e.g., [2, 3, 4]).

[1] Santurkar, Shibani, et al. "Image synthesis with a single (robust) classifier." *arXiv preprint arXiv:1906.09453* (2019).

[2] Tanaka, Akinori. "Discriminator optimal transport." *arXiv preprint arXiv:1910.06832* (2019).

[3] Che, Tong, et al. "Your GAN is secretly an energy-based model and you should use discriminator driven latent sampling." *arXiv preprint arXiv:2003.06060* (2020).

[4] Ansari, Abdul Fatir, Ming Liang Ang, and Harold Soh. "Refining deep generative models via discriminator gradient flow." *arXiv preprint arXiv:2012.00780* (2020).

**Summary Of The Paper:**

The paper proposes BIGRoC (Boosting Image Generation via a Robust Classifier), a method to refine samples from a base generative model using a robust classifier. In this context, a robust classifier means one that has been trained to be robust to adversarial samples. Given a robust classifier and a base generative model (both trained on the same dataset), the key idea involves using the gradients of the robust classifier to update samples generated from the base generative model in the direction that maximizes the conditional probability of the samples' class. Experiments on the CIFAR-10 dataset have been presented that demonstrate that BIGRoC improves the sample quality from several base generative models in terms of the FID and IS metrics.

**Summary Of The Review:**

The paper explores an interesting idea of using robust classifiers to refine samples from generative models; however, the technical contribution is limited. Several directly relevant methods of sample refinement have neither been discussed nor compared against. Overall, the contribution is not significant enough to warrant acceptance.

---

> ### Author Response · Authors · 2021-11-16
> **Reporting additional experiments on ImageNet, clarifying the relation to prior work, and discussing training overhead and sensitivity to the robust classifier's design**
>
> We would like to thank the reviewer for the review and constructive feedback. Our response is given below.
>
> [Novelty, Significance, and Prior Work]
>
> * Our work and [1] both harness the perceptually aligned gradients (PAG) phenomenon, but there are major differences that we would like to highlight. [1] uses PGD over a given adversarial robust classifier for generating images, reaching 7.5 IS on the CIFAR-10 dataset. While this is relatively good, it is far from SOTA. One may wonder whether this IS performance limit is due to the suggested method or the capability of robust classifiers (RoCs). In contrast, our work builds on any existing generative model, including highly performing ones (capable of synthesizing images of quality significantly higher than 7.5 IS), boosting their results by leveraging the perceptually aligned gradients (PAG) phenomenon of RoC. As such, our work exposes the much stronger force that does exist in RoC in capturing high perceptual quality features and using them to attain the sought improvement. The results obtained in our experiments indicate that even highly performing generators can be further boosted. Thus, our work clearly demonstrates that the capabilities of RoC are far beyond the ones exposed in [1]. In summary, although our work might seem conceptually similar to [1], our novel application sheds light on the perceptually aligned features, captured by adversarial robust.
>
>     [2,3,4] are indeed very relevant references that address a seemingly similar problem, and we will include them in the revised version of the paper – thank you for this comment. These papers propose methods for improving generated images using the discriminator of a GAN. In all three, the strategy is to refine the synthesized images by updating the latent vector from which they were generated, using the supervision of the discriminator.
>
>     Our work offers a much simpler and different way of boosting generated images by an external robust classifier, without any need to have access to the latent vectors generating the images, nor to the GAN’s generators or the discriminators. Thus, these methods have much stricter requirements than our method. BigRoC can operate on standalone images - a setup where none of these methods can operate, and this is a significant innovation. Another major difference is that our algorithm operates on the pixel space and not on the latent space, which is a much more challenging setup. For example, DOT [2] proposes the Target Space Optimal Transport algorithm that operates on the pixels, however, this method was applied only on synthetic datasets and didn’t scale well to real images. Therefore, we claim that although these methods seem similar to BigROC, our method operates on-par and even better than these methods, qualitatively and quantitatively, (we refer the reviewer to our experiments on ImageNet) with much fewer requirements and can even be applied on setups that none of them can operate. In addition, BigRoC is truly model agnostic and can operate on images generated by any model, not only GANs.
>
> * As mentioned correctly, our method requires pretrained adversarial robust classifiers which [2,3,4] do not. However, our method is more generic and has capabilities that the others do not possess (as explained above). In addition, we note that the same robust classifier can be utilized to boost all the generated images from all the generative models, and thus, only a single robust classifier is required. Moreover, there are publicly available pretrained adversarial robust classifiers on the common datasets and thus, there is no need to train such networks at all.
>
> [Empirical Evaluation]
>
> * The reviewer is correct – Testing our method on a higher resolution dataset is important for showing that our method is scalable. We kindly refer the reviewer to our official comment added to the submission, which summarizes our experiments on the ImageNet dataset. We note that these improvements are substantially more significant than the ones demonstrated in [2,3] on ImageNet ([4] did not operate on ImageNet).
>
> * We conducted an experiment to discover the effects of the size of $\epsilon$ that defines the threat model used to train the RoC. Applying our method on WGAN-GP with a robust classifier trained with $\epsilon=0.5$ improved the FID from 22.62 to 16.28, while using a robust classifier trained with $\epsilon = 1$ leads to FID of 16. This indicates that the sensitivity to the values of $\epsilon$ is not significant, similar to the claim in Appendix 4 in [1].
>
> * In our work we did not train any model and only used pretrained ones. However, in general, training an adversarial robust classifier has of course more overhead compared to a regular classifier. For example, if the PGD uses K steps, each batch requires (K+1) backpropagation steps (A common choice of K is 7). Nevertheless, we emphasize that there is no need to train such models from scratch since they are available publicly.

---

> > ### Comment · Reviewer_Ezr2 · 2021-11-19
> > **Thanks for your response!**
> >
> > Thank you for your response!
> >
> > * I agree that demonstrating the empirical benefits of robust classifiers in this setting makes for an interesting study; however, the overall technical contribution of such a study is not significant enough to warrant acceptance at ICLR.
> >
> > * [2, 3, 4] exactly address the problem of sample refinement and all these papers are 1-2 years old. For such a specific problem, overlooking all these baselines is in itself a significant weakness of this paper. Further, the argument that these baseline require discriminators is not convincing because the proposed method also requires an additional robust classifier. Moreover, it is unclear how/why operating in pixel space is beneficial if the end goal is the improved quality of samples.
> >
> > * Thank you for providing new results on the ImageNet dataset.
> >
> > In summary, my impression of this paper has not changed. I think the empirical study is interesting but the overall contribution to the community is not significant.

---

> > > ### Author Response · Authors · 2021-11-20
> > > **Response**
> > >
> > > We thank the reviewer for this response. We agree that [2,3,4] are very related and should have been mentioned in our work, and we intend to add them. However, on the technical side, there are major differences between our work and [2,3,4]: Our approach does not require any additional data besides the generated images, while [2,3,4] require in addition the latent codes for each image and the generator producing them. To emphasize, if [2,3,4] and BigRoC were used as real-world applications, our’s would require from the user only the images for the refinement while [2,3,4] would also require the latent codes, the generator model, and the discriminator. The robust classifier we use is arbitrary and detached from the synthesis networks. One last thing: our approach is not limited to GAN’s and it can easily be incorporated for VAE or diffusion methods.
> > >
> > > The claim regarding the pixel space versus latent space was raised to demonstrate the power of our method. While DOT [2] was initially designed to operate in the pixel space, it did not deliver, and due to this fact their method was redesigned to operate in the latent space, which requires much more (access to latent codes and generators, in addition to the discriminator). It means that an operation on the pixels is more challenging but also has benefits, as it requires less knowledge of the synthesis process. [3,4] follow [1]’s path and operate as well in the latent space. In contrast, the fact that BigRoC operates on the pixel space demonstrates the power of our method, comparing the previous work.
> > >
> > > Referring to the statement that our overall technical contribution is not significant, please allow us to summarize the contributions of our paper:
> > > - We present a new and completely different image refinement algorithm, which operates in a much more realistic setting (no need to access latent codes nor generators)
> > > - Our approach adopts a more challenging setup than previous work, as it operates in the pixel domain, and yet succeeds in boosting visual performance.
> > > - Our method performs better on both small resolution and high-resolution images. Note that synthesized images’ refinement is a real-world application, and thus our advantage on high resolution compared to other methods is significant.
> > > - Our work demonstrates better than previous work [1] the impressive capabilities of robust classifiers.
> > >
> > > Due to all of the above, we do believe that our work is innovative and significantly different from previous work and therefore hope that you reconsider given the above.

---

> > > > ### Comment · Reviewer_Ezr2 · 2021-11-28
> > > > **Comment**
> > > >
> > > > Thanks for your response.
> > > >
> > > > As mentioned previously, I appreciate this study of robust classifiers but it does not make significant technical contribution and/or bring any new insights. In my opinion, the overall contribution is a straightforward extension of prior work [1]. Missing discussion and empirical comparisons with directly-related baselines [2, 3, 4] is a *significant* weakness. I hope that the authors will address these issues.

---

> > > > > ### Author Response · Authors · 2021-11-30
> > > > > **Response**
> > > > >
> > > > > Our response included a very clear explanation of the substantial benefits of our proposed method over those discussed in [2,3,4]. Since we already promised to add such a discussion to the camera-ready version, we do not understand how can we better address this issue.
> > > > >
> > > > > Regarding the resemblance to [1], we provided explications both in the paper and in our previous response. On the applicative side, we propose a novel application of PGD on a robust classifier for image refinement task, that attains SOTA performance on image refinement (see our results on ImageNet, compared to [2,3]). More precisely, BigRoC and [1] harness the well-known perceptually aligned gradients (PAG) phenomenon for completely different tasks. In addition, our approach also sheds light on the true capabilities of robust classifiers. We also intend to add this discussion to the camera-ready version upon acceptance.

---

### Official Review · Reviewer_jEYy · 2021-11-02

**Correctness:** 3
**Technical Novelty And Significance:** 3
**Empirical Novelty And Significance:** 2
**Recommendation:** 5
**Confidence:** 3

**Main Review:**

The proposed method has some nice properties. It is model-agnostic and can be used to improve generative models without re-training. The only requirement is a robust classifier trained on the same dataset as the generative models. As stated in the paper, training a classifier is much easier than training a generator. Even better, the same classifier can help all generative models trained on the dataset. These characteristics are attractive in applications.

There are two weaknesses of the paper. First, the experiments are on the low-resolution CIFAR-10 dataset. Although the method is effective on the CIFAR-10 dataset, it is uncertain how well the proposed method performs on normal quality images. For 32x32 images, it is impossible to check out details. Also, low-resolution images are not very useful in real applications. As classifiers could focus more on high-level features, it is not clear how they help synthesize realistic details for high-resolution images.

Second, as stated in the paper, the method is similar to prior work, Santukar et al. 2019 and Turner et al. 2019. The paper lists three differences: (1) the proposed method builds upon the generated images; (2) the proposed method is model-agnostic; and (3) the proposed method can be performed on every image generated by the generative models without throwing out anyone. Although the technical novelty is not significant, the proposed method has advantages and could be useful in applications.


**Summary Of The Paper:**

This paper proposes a model-agnostic method for improving the quality of images produced by generative models. The method requires a robust classifier trained on the same data source as the generative models. Based on the perceptually aligned gradients phenomenon, the proposed method improves the quality of a generated image by the targeted projected gradient descent method with the help of the robust classifier. Experiments on CIFAR-10 show that the proposed method does improve several generative models both quantitively and qualitatively.

**Summary Of The Review:**

Overall, I like the proposed method because it is model-agnostic, simple, and effective. However, as stated in the previous section, since experiments are only performed on low-resolution images, it is not clear how the proposed method performs on images with ordinary resolutions and quality. Novelty is another potential issue, although I think that the paper has made sufficient contribution given its nice characteristics listed above.

---

> ### Author Response · Authors · 2021-11-16
> **Reporting additional experiments on ImageNet, and clarifying the relation to prior work**
>
> We would like to thank the reviewer for the review and the constructive feedback. We address the given comments below:
>
> * In the submitted manuscript, we experimented only using low-resolution images (CIFAR-10), and the qualitative improvement was shown to be very significant (see Figures 3,4,5,7,8,9,10).  However, as the reviewer correctly suggests, it is unclear from the conducted experiments how our method would perform on higher-resolution images. We kindly refer the reviewer to our official comment added to the submission, which summarizes our experiments on the ImageNet dataset.
>
> * Our method is indeed related to the work reported in [1,2], as we state in the paper. Several key differences make our work innovative.
>     * [1] uses PGD over a given adversarial robust classifier for generating images, reaching 7.5 IS on the CIFAR-10 dataset. While this is relatively good, it is far from SOTA. One may wonder whether this IS performance limit is due to the suggested method or the capability of robust classifiers (RoCs). In contrast, our work builds on any existing generative model, including highly performing ones (capable of synthesizing images of quality significantly higher than 7.5 IS), boosting their results by leveraging the perceptually aligned gradients (PAG) phenomenon of RoC. As such, our work exposes the much stronger force that does exist in RoC in capturing high perceptual quality features and using them to attain the sought improvement. The results obtained in our experiments indicate that even highly performing generators can be further boosted. Thus, our work clearly demonstrates that the capabilities of RoC are far beyond the ones exposed in [1]. In addition, we should add that a close inspection of the visual quality of the reported experiments in [1] reveals a weakness in producing visually pleasing results, while our qualitative results are far better.
>     * [2] also tackles the problem of refining the output of generative models. However, their approach is substantially different – instead of improving the quality of any given generated image, their method serves as a 'selector', purging poor quality images. In contrast, our work modifies each of the generated images in the pixel space in order to refine their perceptual quality.
>
> [1] 	Santurkar, Shibani, et al. “Image Synthesis with a Single (Robust) Classifier”. arXiv preprint arXiv:1906.09453 (2019).
>
> [2] 	Turner, Ryan, et al. “Metropolis-Hastings Generative Adversarial Networks”. arXiv preprint arXiv:1811.11357 (2019).

---

> > ### Comment · Reviewer_jEYy · 2021-11-29
> > **comments**
> >
> > Thanks for adding results of ImageNet with a higher resolution. The quantitative comparison shows significant improvement. Qualitatively, the images become sharper, but the improvement is not as significant as CIFAR-10 results. Overall, the paper proposes an empirical model-agnostic method that can improve generative models without re-training. The method can be useful to some applications. On the other hand, I agree with Reviewer Ezr2 that the technical novelty is somehow limited. Also, the lack of comparisons with related methods is an issue that needs to be addressed. Thus, I keep my score.

---

> > > ### Author Response · Authors · 2021-11-30
> > > **Response**
> > >
> > > Thank you for your comments. The qualitative improvement on ImageNet is subjective, however, the quantitative metrics clearly indicate the substantial improvement attained across different architectures, including BigGAN, which is the current SOTA GAN model. Our results on ImageNet are significantly better than the ones attained by the related methods. This, we believe addresses and answers your initial comments.
> > > Regarding the additional claims, we agree that a comparison with related methods should be done in the initial submission. We provided a comprehensive comparison in our reply to reviewer Ezr2 and promised to add it to the camera-ready version. Therefore, we fail to understand why this should affect our acceptance.

---

### Official Review · Reviewer_v7Qv · 2021-11-03

**Correctness:** 3
**Technical Novelty And Significance:** 2
**Empirical Novelty And Significance:** 2
**Recommendation:** 5
**Confidence:** 3

**Main Review:**


Advantages:
1. The improvement on the CIFAR-10 dataset is significant. And the refined images look much better than baselines.
2. The model can work with both conditional/unconditional generative models.

Weakness:
1. In algorithm 1, it is not clear how to use epsilon. Please add more details.
2. In section 2.2, the robust classifier method optimizes both the input images and the network, but in section 3, it only optimizes the input images. The definition is not aligned.
3. Have the authors normalized the input images? Will the input pixel value between [0,1] or [0,255]? In the supplementary, the epsilon of VAE is 25. if the model uses a very large epsilon value, will the refined image still look realistic?
4. This paper only shows the CIFAR-10 results. Will this method generalize to large-scale and high-resolution datasets, such as ImageNet? Currently, people are interested in high-resolution image generation. For example, BigGan can generate high-resolution images. Please the authors test this post-procession method on these generated high-res images.
5. The projected gradient descent has already been used in some image generation works [1].
6. The FID and IS scores of cGAN-PD and BigGAN in this draft are different from the original papers. Please add more details. Also, missing the definitions of '10K' and '50K'.



[1] Xia, Weihao, et al. "Gan inversion: A survey." arXiv preprint arXiv:2101.05278 (2021).

**Summary Of The Paper:**

This paper presents a post-processing method to improve the GAN results. The post-processing consists of a projected gradient descent step to update the generated image to fit its target class, which works with both conditional/unconditional generative models. For the unconditional generative model, the authors design a de-bias method to force the model to uniformly generate images in each class. The results show this model can significantly improve the FID and IS scores on CIFAR-10 dataset. Additionally, the authors show this post-processing also works for image interpolation.


**Summary Of The Review:**

Given the results and novelty are marginal, I think this paper is between boardline. Please the authors address my questions. I am happy to change my score.

---

> ### Author Response · Authors · 2021-11-16
> **Reporting additional experiments on ImageNet; Discussing relation to other work that uses PGD; Clarifications regarding several topics (the role of epsilon, a gap between section 2.2 and 3, pixel value normalization, FID and IS results versus reported ones in the literature)**
>
> We would like to thank the reviewer for the review and constructive feedback. Our response is given below:
>
> 1. $\epsilon$ is the value of the maximal perturbation allowed by the threat model $\Delta$. Thus, it is used to project the adversarial noise $\delta$ onto the threat model. In the $l_2$ case, this projection is given by $\Pi_{\epsilon}(\delta)=\epsilon * (\frac{\delta}{max[\epsilon, ||\delta||_2]})$.
>
> 2. Sections 2.2 and 3 do not describe the same method. In section 2.2, we describe the training process of an adversarial robust classifier, which indeed requires optimization both on the input images and the network’s parameters. However, in section 3 we present our method, which utilizes a pretrained adversarial robust classifier to refine the generated images. To this end, only the input images are optimized.
>
> 3. Our inputs are normalized to the range of [-1,1]. $\epsilon = 25$ corresponds to an average pixel change of 0.45 in resolution of $32\times32$ ($\sqrt(\frac{25^2}{32\times32\times3}) = 0.45$), which is equivalent to an average pixel change of 57.51, in terms of values in the range [0,255]. This perturbation is indeed large, however, the perceptual quality of the synthesized images by the VAE is quite poor, and requires a significant modification (and thus usage of a large $\epsilon$). As can be seen in Figure 8 in the appendix, the refined images look much more realistic than the ones generated by a VAE.
>
> 4. Thank you for this comment - we kindly refer the reviewer to our official comment added to the submission, which summarizes our experiments on the ImageNet dataset.
>
> 5. Projected Gradient Descent (PGD) is a generic optimization strategy used in various contexts, in which there is a need to enforce a constraint through a projection. In [1] PGD is deployed for solving inverse problems via an existing (pretrained) generator, where the projection is on the range of the generator. This is substantially different from our use of PGD, in which we boost generated images of any generative model, while projecting onto a sphere constraint of allowed perturbation. Our application of the PGD over a robust classifier to refine generated images is novel and unique. In addition, it demonstrates the remarkable perceptual quality features captured by adversarial robust classifiers, far beyond prior work.
>
> 6. In this work we did not train any model and used only available pretrained ones from verified sources. Also, we used common methods to evaluate the IS and FID metrics. Our cGAN-PD pretrained model was taken from mimicry [2] repository, aimed towards the reproducibility of GAN research, and our results are aligned with the ones reported in mimicry. Our BigGAN was taken from the Differentiable Augmentation for Data-Efficient GAN Training [3] paper’s GitHub repository. We have used these and other pretrained models to generate sets of images and used our method to refine them. Then, we evaluated the FID and IS on the generated images and the boosted ones. \
> Thank you for your remark regarding the 10K and the 50K – we will add a brief explanation in the paper. The 10K and the 50K corresponds to the number of synthesized samples used for evaluating the FID metric. In the 10K case, the generated images are compared to the ones of the test set, where the 50K synthesized images are compared to the ones of the training set. These two evaluation procedures are common, and we decided to use them both.
>
>
> [1] 	Xia, Weihao, et al. "Gan inversion: A survey". arXiv preprint arXiv:2101.05278 (2021).
>
> [2] 	Sin Lee, Kwot, et al. “Mimicry: Towards the Reproducibility of GAN Research”. arXiv preprint arXiv:2005.0249 (2020). https://github.com/kwotsin/mimicry
>
> [3] 	Zhao, Shengyu, et al. “Differentiable Augmentation for Data-Efficient GAN Training”. arXiv preprint arXiv: 2006.10738 (2020).

---

> > ### Comment · Reviewer_v7Qv · 2021-11-29
> > **Responce**
> >
> > Thanks the authors for addressing my questions and adding the testing results on ImageNet dataset. The improvement is significantly. However, I agree with the other two reviewers, the contribution is limited and this paper is missing the empirical comparisons with [2,3,4]. I will downgrade my score to 5.

---

> > > ### Author Response · Authors · 2021-12-02
> > > **Response**
> > >
> > > Thank you for your response. We would like to refer the reviewer to our comprehensive comparison with [2,3,4] in our responses to reviewer Ezr2. To summarize, our method is simpler, operates better than the aforementioned methods with fewer requirements, and is capable of operating in a setting that the laters simply cannot. As promised in our previous responses, we aim to add a comparison to [2,3,4] to the camera-ready version upon acceptance.

---

### Author Response · Authors · 2021-11-16
**Reporting additional experiments on ImageNet**

Indeed, as all the reviewers correctly commented, testing our method on a higher resolution dataset is important for showing that our method is scalable. We have conducted additional experiments on the ImageNet dataset ($128\times128$). We took a pretrained robust classifier from [1] and the publicly available pre-trained unconditional GANs - SNGAN, SSGAN, and InfoMaxGAN - from mimicry [2] and evaluated our method using FID and Inception Score. We used $\epsilon = 40, T = 60, \alpha = 1$ to boost these models. Our method leads to a significant improvement in both metrics, shown in the following format (Baseline FID – Our FID | Baseline IS – Our IS):

SNGAN:           68.28 - 40.40 | 13.05 - 71.67

SSGAN:           63.60 – 38.93 | 13.75 – 73.94

InfoMaxGAN:  60.61 – 37.70 | 13.79 – 75.49

The improvement in the Inception Score is partly due to the fact that these models are unconditional and our debiasing mechanism leads to more uniform labels’ distribution, which the IS favors. The significant improvement in the FID indicates the higher fidelity of the images produced by our method. In addition to these models, we experimented with a pretrained BigGAN (from hugging face GitHub), using $\epsilon = 10, T = 60, \alpha = 0.25$. We note that this model is conditional and thus our debiasing mechanism is not activated. We use truncations of 0.1, 0.5, 1 and report the following results (Baseline FID – Our FID | Baseline IS – Our IS).

BigGAN (truncation 1.0):    5.88 - 5.65      | 108.74 – 124.54

BigGAN (truncation 0.5): 15.84 – 13.17 | 164.88 – 239.63

BigGAN (truncation 0.1): 28.88 – 23.49 | 179.23 – 245.66

As can be seen from these experiments, our method is effective also on high-resolution images, and capable of substantially improving the SOTA GAN. In addition to the quantitative results on ImageNet, we demonstrate the qualitative improvement attain by our method in the following link:
https://drive.google.com/drive/folders/1tBM4hlBJDtB1SH1zJxAWO9Iar3bZwjBa?usp=sharing. We thank the reviewers for pointing this issue, and we plan on adding the results of these experiments to the revised version of the paper.


[1] Robustness (Python Library). https://github.com/MadryLab/robustness.

[2] Sin Lee, Kwot, et al. “Mimicry: Towards the Reproducibility of GAN Research”. arXiv preprint arXiv:2005.0249 (2020). https://github.com/kwotsin/mimicry

---

### Decision · Program_Chairs · 2022-01-20

**Decision:**

Reject

**Comment:**

The paper proposes to improve generated images via a post-processing update procedure guided by gradients from a robust classifier.  After the author response and discussion, all reviewers agree that the paper is below the acceptance threshold.  Reviewer concerns include limited technical novelty and missing experimental comparison to relevant baselines.